# Therapeutic and Preventive Interventions in Adolescents with Borderline Personality Disorder: Recent Findings, Current Challenges, and Future Directions

**DOI:** 10.3390/jcm12206668

**Published:** 2023-10-21

**Authors:** Nadège Bourvis, David Cohen, Xavier Benarous

**Affiliations:** 1Centre Hospitalier Intercommunal Toulon la Seyne (CHITS), 83000 Toulon, France; 2Maison des Adolescents du Var, 83000 Toulon, France; 3Service Universitaire de Psychiatrie Infanto Juvenile, Aix-Marseille Université, 13009 Marseille, France; 4Institut des Systèmes Intelligents et Robotique, APHP-Sorbonne Université, 75651 Paris, France; david.cohen@aphp.fr (D.C.); xavier.benarous@aphp.fr (X.B.); 5GH Pitié-Salpêtrière, 75013 Paris, France

**Keywords:** borderline personality disorder, adolescence, structured therapy

## Abstract

Background: Borderline personality disorder (BPD) has long suffered from overshadowing in adolescents and hopelessness from the psychiatrists themselves. Comprehensive guidelines for this age group are lacking. Aims: This narrative review aims to describe current recommendations for BPD and recent empirical evidence on effective treatments (both pharmacological and non-pharmacological) and preventive approaches. Innovative approaches, based on recent and original research on BPD adolescents, are also discussed. Results: Very low-certainty evidence has supported that medication has a positive effect on core BPD symptoms in adolescents. Medication prescribed for suicidal crises or associated disorders should be included in a global therapeutic plan, including efficacy reassessment, treatment duration, and a security plan. The overall benefit of structured psychotherapy for adolescents with BPD (cognitive behavioral therapy, mentalization-based therapy, dialectic behavioral therapy, and group therapy) is more important for self-harm behaviors than other BPD symptoms. Their specific efficacy, although difficult to distinguish from the overall non-specific effect of integrative care. Conclusions: structured care of young BPD individuals should be based on the following principles: (1) setting the frame of care, including recognition of the diagnosis, and sharing information with patients and families about symptoms, prognosis, and putative psychological mechanisms involved; and (2) promoting comprehensive approaches, including both specific and non specific therapy, ecological interventions, community care, and preventive interventions in at-risk groups.

## 1. Introduction

Borderline Personality Disorder (BPD) is a personality disorder characterized by severe emotional dysregulation, impulsive acts, suicidal and self-harm behaviors, as well as disturbed interpersonal relationships, a chronic fear of abandonment, and a recurrent feeling of emptiness [1]. Substantial levels of impairment are associated with BPD, with, in particular, a high risk of death by suicide and poor personal and social functioning.

The individual and social burden associated with BPD is expected to be particularly high for adolescent-onset forms [2,3]. In this context, emotional and behavioral dysregulation may durably affect co-occurring socio-emotional development, in particular social identity or socialization abilities [4]. Symptoms can also interfere with academic functioning with long-term influence in terms of socio-economic status [5].

The prevalence of BPD in adolescents in community-based samples has been reported to be around 2–3%. Adolescent-onset BPD tends to be especially represented in clinical samples, with a frequency gradient according to patients’ level of severity [6]. The rate of BPD in adolescents referred to outpatient facilities was estimated at 11%, and between 19 and 53% in inpatient facilities. The rate of BPD tended to be dramatically high in adolescents referred for full-time hospitalization or emergency services for suicidal behaviors [7]. Register-based studies also showed that the diagnosis of BPD in adolescents has shown a steady increase over the last decades [8,9].

Difficulties in the recognition of adolescent forms of BPD stem from multiple causes. The diagnosis of personality disorder in adolescents had long been a matter of debate among mental health practitioners before it became progressively consensual in clinical communities and supported by national and international societies of child and adolescent psychiatry [4]. Empirical data, such as clinical trials on the effectiveness of interventions, on the adolescent form of BPD are scarce, and clinical practices have been largely influenced by findings from the adult population. As a result, clinical practices widely differ among teams or countries. Due to the features of the disorder, care is often solicited by patients and families in the context of an emergency (e.g., after suicidal attempts), with a cost-effectiveness on general health and well-being that remains particularly low.

Overall, there is now broad evidence-based consensus that BPD is a reliable, valid, and common mental disorder that begins in adolescence with severe clinical, functional, and socio-economic consequences. Early recognition of the diagnosis and appropriate care seem crucial for a better prognosis, although delay in diagnosis and treatment remains the norm.

In this review, we aim to present the current understanding of BPD in adolescents and how this information may be used in the strategy of care. We expose the current available evidence for the principal specific treatments, i.e., pharmacotherapy, psychotherapy, and prevention. We finally discuss what seems to work and what future directions to explore in relation to the current research in BPD.

## 2. Methods

The purpose of this paper is to provide up-to-date directions for managing BPD in adolescents. We aimed to review the empirical evidence supporting the benefits in children and adolescents with BPD of (1) pharmacological treatment, (2) psychotherapeutic interventions, and (3) preventive approaches. In the last section, we present the results of the most recent research on BPD and adolescence and how it highlights new directions for more tailored care. Considering the paucity of data for each intervention, the mean level of evidence of reviewed studies, and the diversity of the clinical constructs mentioned, proceeding with a systematic review or meta-analysis seemed ill-adapted.

We have proceeded to a detailed narrative review of articles published on BPD in children and adolescents. Relevant articles were obtained through Cochrane and PubMed. Each database was searched from 1980 to 2023. In addition, we hand-searched reference lists of identified articles and pertinent reviews for additional studies. We used the following search terms: “Borderline Personality Disorder” OR “emotionally personality disorder”. All studies were reviewed if they examined the effect of an intervention in children or adolescents with BPD, either therapeutic or preventive. Findings from RCTs and the most relevant case-controlled studies are presented in dedicated tables. The full texts of the articles were reviewed to examine the population, the intervention, and the main outcomes. We categorized the overall level of evidence presented for each intervention using the United States Preventive Services Task Force (USPSTF) criteria: level I evidence for at least one well-designed RCT supporting a treatment’s possible efficacy; level II-1 for a well-designed controlled trial without randomization; level II-2 for at least one well-designed cohort or case–control study; and level II-3 for multiple time series design. We decided to present only studies with a higher level of evidence than USPS’s level II-2. Results were then critically discussed and put into perspective based on previously published guidelines and using a developmental and ecological framework for the emergence of BPD in adolescence. Areas of improvement for further clinical research are then detailed.

## 3. Results

### 3.1. Pharmacological Treatments

Polypharmacy is frequently observed in adolescents with BPD [2,10,11], although very low-certainty evidence has supported that medication has a positive effect on core BPD symptoms [11,12]. A recent Cochrane meta-analysis conducted on subjects with BPD (both adolescents and adults) found that, compared with placebo, medication has little to no effect on BPD symptom severity. The Standardized Mean Difference (SMD) of BPD symptoms for antipsychotics compared to placebo was SMD −0.18, 95% CI: −0.45 to 0.08; antidepressants: SMD −0.27, 95% CI: −0.65 to 1.18; and mood stabilizers: SMD −0.07, 95% CI: −0.43 to 0.57 [12]. The evidence was also very uncertain about the effect of medication on self-harm, suicide-related outcomes, and psychosocial functioning compared with placebo.

Only one RCT was conducted on participants with BPD below the age of 18 [13]. In this study, Amminger et al. (2013) conducted a post hoc subgroup analysis of a double-blind RCT on just 15 adolescents with BPD (mean age 16.2 years, [SD 2.1]), and they showed a positive effect of a 12 week fatty acid supplementation (long-chain omega-3 polyunsaturated fatty acids) on BPD symptoms.

To date, no medications have been approved by national regulatory agencies for the treatment of BPD. In practice, psychotropic drugs are not recommended in the treatment of BPD in adolescents [14], as reviewed by the Australian National Health and Medical Research Council (NHMRC) [15] and the UK National Institute for Health and Care Excellence (NICE) [16]. According to these guidelines, if a pharmacological treatment must be used, second-generation antipsychotics should be prioritized in addition to other therapies, for example, in the context of a suicidal crisis [10]. Medication prescribed for suicidal crisis should be included in a global therapeutic plan, considering their accessibility, their lethality, possible interactions with other drugs, alcohol, and substances, the need to inform other professionals, the planned duration, and the withdrawal strategy [10,14]. The emergence of acute psychotic symptoms is not an indication of prolonged antipsychotic treatment. Evidence supporting the use of antidepressants, e.g., selective serotonin reuptake inhibitors (SSRIs), on emotional BPD symptoms (e.g., chronic anger, irritability, or affective lability), is very low in general and particularly in adolescents. Benzodiazepines should not be prescribed to youths with BPD, considering the risk of substance dependence [10]. In adults, low-certainty evidence supports the benefit of pharmacological treatments specifically on BPD symptoms, while second-generation antipsychotics were regarded as possibly effective on associated general psychiatric symptoms in these patients [11].

However, adolescents with BPD could require medication to treat co-occurring psychiatric or neurodevelopmental disorders. SSRI will be indicated in cases of severe and/or resistant major depressive disorder in adolescents with BPD, as it is in the rest of the general population. Psychostimulants should be prescribed for authentic and well-documented attention deficit disorder [17]. Melatonin (either a standard formulation or prolonged release) has been shown to be of interest in the case of severe sleep disturbance/insomnia in adolescents. A recent population-based study found a decreased relative risk for self-harm (relative risk: 0.58, 95% CI: 0.46 to 0.73) in the month following treatment initiation among adolescents with psychiatric disorders [18]. The use of sedative-hypnotics for sleep difficulties or opioid-agonists/antagonists for self-harm behavior is not recommended.

### 3.2. Psychotherapeutic Treatment

Psychotherapeutic interventions are the first-line treatment for adolescents with BPD, according to expert authors [2] and national guidelines (Australian NHMRC, Canberra, Australia [15] and UK NICE, London, UK [16]).

A Cochrane review showed that the overall effect of psychotherapeutic treatments for BPD was lower in adolescents compared to adults [19] (respectively, mean effect size, ES = 0.45 vs. ES = 0.85). To date, ten RCTs have been conducted to determine the effectiveness of psychotherapeutic treatments in adolescents with BPD (Table 1). The overall benefit of these interventions seems more important for self-harm immediately than at the end of the intervention, while the effect after a few months of follow-up remains difficult to differentiate from the non-specific impact of Treatment As Usual (TAU).

#### 3.2.1. Supportive Psychotherapy (SP)

SP is the most widely used psychological treatment for adolescents with BPD and may be initiated without a confirmed diagnosis [20]. It is based on a nonjudgmental, empathizing approach. The professionals encourage the formulation of the potential problems encountered with a specific focus on self-esteem and promoting adaptive coping strategies. This approach is non-structured and non-specific. It is usually based on individual sessions with varied frequency on a weekly basis. It has been used in RCT as an active control group [21].

#### 3.2.2. Cognitive Behavioral Therapy (CBT)

Principles of CBT developed for depressive and anxiety disorders have been applied to subjects with BPD. Cognitive Analytic Therapy (CAT) has been developed as a psychotherapy for BPD, focusing on problematic self-management, interpersonal relationship patterns, and associated thoughts, emotions, and behavioral responses [22]. Its application to youth with BPD has been described by McCutcheon [23]. Chanen [24] compared the effect of 24 weekly sessions of CBT-type CAT provided to 41 adolescents with BPD traits to that of 37 patients receiving good clinical care. After 2 years, no difference was found between the two groups. More recently, the same team did not find a significant difference in terms of clinical efficacy between Enhanced Usual Care (EUC) + CAT and EUC + befriending [25]. Schuppert [26,27] failed to find an effect of 19 sessions of Emotion Regulation Therapy (ERT) on BPD symptoms assessed at 4 and 6 month follow-ups compared to TAU. ERT is a manualized treatment that integrates components of cognitive behavior with a specific focus on emotional awareness.

#### 3.2.3. Mentalization-Based Therapy (MBT)

MBT is a therapy specifically designed for individuals with BPD, developed and manualized by Fonagy and Bateman [28]. This therapy is inspired by attachment theory, in particular the internal working model of attachment [29]. Poor mentalizing is regarded as a core feature of BPD [30]. The treatment aims to increase a patient’s mental capacity, which would result in increased affect regulation, attention, self-control, interpersonal competencies, and reduced self-harm. MBT takes place twice per week, with sessions alternating between group therapy and individual treatment. An adaptation for adolescents was developed with a shorter duration (5 month structured treatment approach including individual therapy, combined psychotherapy with the individual therapist also being one of the group therapists, and closed-group therapy to enhance cohesion and a feeling of security) [31]. Roussouw [32] compared the effect of 12 months of MBT versus TAU in 80 adolescents aged 12–17 years old with borderline symptoms. A decrease in self-reported self-harm, depression, and borderline symptoms was higher in those with MBT compared to TAU, while no difference was reported with regards to risk-taking. Beck [33] found no significant difference between adolescents with BPD receiving MBT and those receiving TAU at the end of treatment on primary and secondary outcomes.

#### 3.2.4. Dialectic Behavioral Therapy (DBT)

DBT has been developed by Linehan [34,35] as a multifaceted psychotherapeutic treatment for BPD, including cognitive behavior training, mindfulness meditation, behaviorism, and dialectics approaches. Compared to traditional CBT, DBT places specific attention on the validation of a client’s emotional experiences and the acceptance of negative emotions. It aims to prevent reinforcement between emotional avoidance and self-harm behavior. It encompasses four modules: mindfulness, emotion regulation, interpersonal effectiveness, and distress tolerance, and usually takes place in three weekly, multimodal therapeutic sessions. DBT has been adapted to the adolescent population (DBT-A) [35,36,37]. In addition to individual psychotherapy, DBT-A aimed to provide several family-focused interventions (e.g., family meetings, telephone coaching for patients and family members, and multifamily training groups). The supervision of the therapist is also part of the requirements for DBT-A. Recently, DBT has been further adapted with a mobile app [38]. Three RCTs using active placebo control have supported the benefit of using DBT in adolescents in its initial form [21] or in the adolescent-adapted version [39,40] (Table 1). The benefit was observed on self-harm and suicidal behaviors rather than on other BPD symptoms, with the effect persisting six months and then twelve months after the end of the intervention [39].

#### 3.2.5. Group Therapy

Group therapy has also been used for adolescents with BPD [41]. For adolescents, it has been argued that group cohesion is very important and related, even more so than for adults, to achieving positive outcomes for a range of different conditions [42]. Group interventions confer many potential advantages, such as cost, time, and therapist resources, as well as the advantage of allowing adolescents to work alongside peers with comparable problems. Group settings could also enhance the risk of teenagers influencing each other in a negative way because of the increased focus on negative emotions, which significantly lowers thresholds for emotional reactivity and arousal. A meta-analysis of 24 studies conducted mainly in adults showed the superiority of group therapy compared to TAU for BPD patients on a range of different measures, including BPD symptoms, suicidality, depression, anxiety, and general mental health [43].

### 3.3. Preventive Approaches

Preventive approaches can be divided into two categories: primary prevention, which focuses on preventing the onset of the disease, and secondary prevention, which focuses on early detection and interventions. Tertiary prevention refers to the management of the disease already diagnosed to prevent the development of complications. This last aspect has been discussed earlier in the current paper.

#### 3.3.1. Secondary Prevention

Several programs of secondary prevention have been developed for adolescents with BPD. As developed by Chanen et al., their common principle is to prioritize early interventions, even when BPD cannot be fully diagnosed with regards to the current criteria. As an example, the HYPE clinic program (Helping Young People Early) is a specific service model designed to take care of adolescents at risk for BPD. The inclusion cut-off is set at three criteria (or two criteria plus a risk factor) [44] to be enrolled in the program. A randomized clinical trial including 139 adolescents with BPD compared three forms of early interventions over 12 months: HYPE + specific therapy, HYPE + befriending therapy, or a general youth mental health service + befriending therapy [25]. As a result, early intervention was effective, did require specific youth-oriented clinical case management and psychiatric care, but was not dependent on specific individual psychotherapy [25]. As reviewed by the Australian NHMRC [15] and NICE in the UK [16], preventive programs are generally recommended for BPD.

#### 3.3.2. Primary Prevention

As mentioned above, primary prevention aims to reduce the prevalence of the disease in the general population. A relevant strategy may consist of implementing specific interventions in targeted groups that appear to be at-risk for developing the disease, knowing that the main identified risk factor is to have been through Adverse Childhood Events (ACEs). Of note, for this type of high level of complexity, where the symptoms appear more than a decade after exposure, the methodological caveats are numerous. We present here three approaches that address this issue of prevention in different at-risk groups, all of them aiming to improve the early life experience:Parenting of children with behavioral disturbances: A pilot study was recently published by the team of Muratori et al. [45], showing that mindful parenting interventions could help reduce the additional risk factors needed to develop future BPD in children already presenting behavioral disturbances. Mindful parenting promotes increasing parenting skills by enhancing attention and self-regulation skills in parents, helping them to pay attention, in a non-judgmental way, to the present moment and to their interaction with their child. In a sample of 16 mothers of children with behavioral disorders, they found significant positive changes in no-reaction skills and a significant decrease in negative parenting practices;Early Mother-Infant Bonding: the team of Fowler [46] addressed the association between early adverse parenting experiences and BPD. This 14 year longitudinal study on 64 mother–child dyads focused on Maternal Bonding Impairment (MBI) during the 2 weeks postpartum only and its interactions with child temperament (at age 5) and child sex as predictors of BPD symptoms and general personality dysfunction in adolescence. It showed that higher MBI was a significant predictor of general personality dysfunction as defined in Criterion A of the DSM-5. Also, BPD symptoms in girls but not boys were dependent on maternal bonding. These results indicate that children at risk of developing personality pathology can be identified early in life and may help target at-risk dyads for selective early prevention;Mothers with BPD: Another even more specific at-risk group for developing BPD are the children of mothers diagnosed with BPD. The German ProChild study will aim, in a controlled trial, to compare the effect of parenting interventions vs. no intervention in children aged 6 months to 6 years from mothers with a diagnosis of BPD [47]. The primary outcomes are changes in parenting from baseline to post-intervention and follow-up at 6 and 9 months after intervention, thus not reflecting the occurrence of BPD in future children but may provide some data on the effectiveness of such interventions.

Also, past exposure to ACEs is known to be particularly high in children and adolescents in foster care, who are also more at-risk to develop BPD symptoms. This is associated with more internalizing and externalizing problems, which, in turn, leads to more parenting stress in foster families [48] or by other professionals. A large meta-analysis of longitudinal research on the development of foster care children found that foster care does not negatively or positively affect foster children’s developmental trajectories, which, given the initial prognosis, is not a good result [49]. To our knowledge, no specific program has been evaluated, but research needs to be developed in this field [50].

**Table 1 jcm-12-06668-t001:** RCTs on psychotherapeutic treatment in BPD (subsyndromal and full-blown disorder) in children and adolescents (based on [51,52]).

Study	BPD Inclusion Criteria	Population/Setting	Assessment	Psychotherapic Intervention	Main Outcomes
Chanen (2008) [24] Australia	≥2 DSM-5 BPD criteria and risk factors	N = 78 76% female Age = 16.4 (0.9) (15–18) RCT with a 24 month FU CBT/CAT (n = 41) TAU (n = 37)	Symptoms: SCID-II, YSR Functioning SOFAS	CBT/CAT Duration: 24 weekly sessions	No difference between groups at FU CBT/CAT improved more rapidly
Schuppert (2009) [26] The Netherlands	At least mood instability criterion of DSM-IV BPD and one more criterion	N = 78 88% female Age = 16.1 (1.2) (14–19) RCT with a 4 month FU CBT/ERT (n = 23) TAU (n = 20)	Symptoms: BPDSI IV; YSR Functioning SOFAS	CBT/ERT Duration: 17 weekly Sessions + 2 booster sessions	No difference between groups on BPD symptoms at FU Subjects with CBT-ERT had improved internal locus of control and greater control over mood swings
Schuppert (2012) [27] The Netherlands	≥2 DSM-IV BPD criteria and risk factors	N = 109 96% female Age = 16.0 (1.2) (14–19) RCT with a 6 month FU CBT/ERT (n = 54) TAU (n = 55)	Symptoms: BPDSI, SCL-90-R Functioning YQOL	CBT/ERT Duration: 17 weekly Sessions + 2 booster sessions	No difference between groups on BPD symptoms at FU
Roussouw (2012) [32] UK	≥1 episode of self-harm within the past month + ≥2 DSM-IV BPD symptom criteria	N = 80 85% female Age = 14.7 (NA) (12–17) RCT with a 12 month FU MBT (n = 40) TAU (n = 40)	Symptoms: BPFS-C, MFQ, and RTSHI Functioning	MBT Duration: 12 months	MBT-A was superior to TAU in reducing self-reported self-harm, depression, and borderline symptoms. At 12 months, 58% of the TAU group met the CI-BPD criteria, but only 33% of the MBT-A group met the criteria (*p* < 0.05). No difference in risk-taking
Mehlum (2014, 2016) [39,40] Norway	≥2 episode of self-harm; one within the last 16 weeks + ≥2 DSM-IV BPD symptom criteria	N = 77 % female NA Age = 15.6 (1.5) (12–18) RCT with a 4 month (2014) and 12 month FU (2016) DBT-A (n = 39) EUC (n = 38)	Symptoms: BSL, MADRS, Parasuicidal behavior Functioning CGAS, Hospital/ER visits	DBT-A Duration: 19 weekly sessions	At 4 months, reductions in self-harm, suicidal ideation, and depressive symptoms were higher in subjects with DBT-A compared to EUC. No difference in BPD symptoms. At 12 months, reductions in self-harm were higher in subjects with DBT-A compared to EUC over the FU period. No other differences in other outcomes were significant
McCauley (2018) [21] USA	Prior lifetime suicide attempt + ≥3 DSM-5 BPD criteria	N = 137 95% female Age = 17.9 (1.5) (12–18) RCT with a 12 month FU DBT (n = 72) SP (n = 65)	Symptoms: SASII SIQ	DBT Duration: 6 months	DBT was superior to SP for reducing suicide attempts, NSSI, and self-harm at 6 months. No difference at FU (12 months).
Beck (2020) [33] Denmark	≥4 DSM-5 BPD symptom criteria	N = 112 99% female Age = 15.8 (1.1) RCT with a 12 month FU	Symptoms: BPFS-C, RTSHI Functioning: CGAS	MBT Duration: 12 months	No difference between groups at the end of treatment in primary and secondary outcomes 29% of the MBT group completed less than half of the sessions, compared with 7% of the control group
Chanen (2022) [25] Australia	DSM-IV BPD criteria	N = 139 81% female Age = 19.1 (2.8) (15–25) RCT with 12 month FU EUC (HYPE) + CAT (n = 40) EUC (HYPE) + befriending (n = 45) TAU + befriending (n = 43)	Symptoms: IIP-CV Functioning: SAS-SR	EUC: Helping Young People Early (HYPE) dedicated BPD service. Duration: 12 months	No difference between groups on the 12 month primary endpoints EUC + SP > TAU + SP was superior for treatment attendance and completion

Note: Abbreviations that have not been defined in the text. BDI: Beck Depression Inventory; BHS: Beck Hopelessness Scale; BPDSI: Borderline Personality Disorder Severity Index-IV; BPFS-C: Borderline personality features scale for children; BSL: borderline symptom list; CGAS: Children’s Global Assessment Scale; DPS: Diagnostic Interview Schedule for Children-Predictive Scales; HoNOSCA-F: Health of the Nation Outcomes Scales for Children and Adolescents; IIP-CV: Inventory of Interpersonal Problems Circumplex Version; MADRS: Montgomery–Asberg Depression Rating Scale; MACI: Millon Adolescent Clinical Inventory; MFQ: Mood and Feelings Questionnaire; RSIQ: Reynold Suicidal Ideation Questionnaire; RTSHI: Risk-Taking and Self-Harm Inventory; SASII: suicide attempt self-injury interview; SAS-SR: Social Adjustment Scale Self-report; SCID-II: structured clinical interview for DSM-IV axis II disorders; SCL-90-R: The symptom check-list; SBFT: Systemic Behavioral Family Therapy; SBT: Skill-Based Treatment; SIQ: Suicidal Ideation Questionnaire; SOFAS: Social and Occupational Functioning Assessment Scale; Youth Quality of Life Instrument; YSR: Youth Self-Report Questionnaire.

## 4. Discussion

BPD in adolescents has progressively become an active field in child and adolescent mental health over the last decades. We discussed how recent findings on the cognitive and neurobiological impairments underpinning some features of the disorder could help develop new therapeutic targets.

### 4.1. A Developmental Model of BPD and Treatment Planning

Bourvis et al. [53] have completed Linehan’s biopsychosocial model of the emergence of BPD to better account for developmental factors and, in particular, the role of early trauma [34]. Figure 1 shows the principal aspects of this model: the risk factors are divided into two categories that are firstly related to the environment (early trauma, invalidating affective environment), labeled as childhood adversity, and, secondly, associated with individual features (temperamental or cognitive vulnerability). These risk factors may constitute the root of the chronic features of the disease, i.e., a vulnerable sense of self (with a chronic feeling of void, of loneliness, a sense of helplessness and depressive mood, a deficit of insight), and unstable interpersonal relationships, underpinned by an intense desire to be loved and the fear of abandonment. The “acute symptoms” emerge when this already unstable functioning is overwhelmed, which may happen secondary to an apparent minor stimulus (relational delusion, frustration, etc.). The external stress cannot be addressed using individual psychological resources, leading to intense emotional distress that is finally acted out. The rational and symbolic functions may also be overwhelmed, leading to transitory psychotic experiences.

Other psychiatric disorders may be intertwined with BPD symptoms. Indeed, a vast number of patients may have received several diagnoses (e.g., depression, substance use disorder, ADHD, etc.) before a more global assessment of BPD is made. In fact, very few patients with a BPD diagnosis fail to meet criteria for another psychiatric diagnosis. This comprehensive model may help patients, families, and non-medical professionals understand why the clinical presentation may be so labile while remaining consistent with the diagnosis. Disorders with the highest rates of co-occurrence with BPD are mood disorders, anxiety disorders, substance use disorders, and other personality disorders.

The narrative supported by the model may find an echo in the patients and relatives experiences. This may help set a frame for future care and possibly the contractualization of treatment planning, ultimately reducing the risk of care discontinuity [2,54]. Ethical concerns about the treatment of BPD in adolescents are also worth sharing with patients, families, and professionals. Informed consent and patient confidentiality should be clearly guaranteed and stated to promote long-term treatment adhesion. The potential for stigmatization should be explained to the patient and family. Some data about the prognosis of BPD is also useful to share with families. Of note, suicidality, as well as the most acute complications due to impulsivity and affective dysregulation, tend to decrease with time [55,56]. The reduction in the level of BPD symptoms may lead to remission when BPD is modeled as a categorical diagnosis [57]. Nevertheless, research shows that symptoms tend to switch to more chronic issues such as maladaptive interpersonal functioning and other functional impairments that are better assessed with dimensional models and support the need for maintaining professional care [58]. Overall, better knowledge of BPD in terms of developmental trajectory helps improve the understanding and treatment adhesion of BPD adolescents and their families.

### 4.2. The Embodiment of Early Adversity and Future Therapeutic Targets

Real-time recordings show that BPD adolescents exhibit an increased stress reaction when compared to controls [59]. This is in line with previous findings in adult BPD patients (e.g., impaired cardiovascular autonomic modulation; lower resting state vagal tone) [53,60]. The dysfunction of the Hypothalamic–Pituitary–Adrenal (HPA), the main support of the stress system, appears to be associated with the experience of early adversity [61,62].

The neural correlates of this enhanced stress response in BPD subjects have been summarized by Gawda et al. [63]. As mentioned earlier, female subjects with BPD and a personal history of abuse in childhood show hyper-reactivity of the HPA axis [64], while hyper-reactivity of the autonomic nervous system is also shown in BPD women with a history of sexual or physical abuse [65]. Furthermore, on a structural level, brain variations such as smaller amygdala, hippocampus, anterior cingulate, and orbitofrontal cortex volumes have been found in people with BPD [63]. As reviewed in [66], functional neuroimaging data suggest that the underlying neural substrate involves hyperactivation in the amygdala to affective facial stimuli and altered activation in the anterior cingulate, inferior frontal gyrus, and superior temporal sulcus, particularly during social emotion processing tasks. These functional abnormalities could result in heightened sensitivity to emotional cues in interpersonal scenarios, self-referential emotional processing, and dysregulated emotional processing.

But how does this early adverse event find a physical embodiment; in other words, “how does adversity gets under the skin?” [67]. In animal models (rats), the effects of early adverse life experiences were shown to be mediated through the epigenetic regulation of hippocampal glucocorticoid receptor expression [67]. In mice, similar early epigenetic changes of the nucleic acids coding for proteins involved in the HPA axis were shown to be associated with early life adversity [68]. In humans, the analysis of postmortem hippocampus obtained from suicide victims with a history of childhood abuse showed decreased levels of glucocorticoid receptor mRNAs and similar epigenetic modifications, suggesting that the same mechanisms may occur [69].

Overall, various evidence illustrates that the relationship between early adverse experiences and BPD symptoms may be explained by the effects of early trauma on the developing and highly plastic infantile brain (epigenetic modulation, alteration of the HPA axis, and persistent changes in stress reactivity). To work on stress responsivity, body-centered approaches that focus on neurovegetative regulation, such as cardiac coherence or other breathing techniques, should be considered and properly assessed. Patients suffering from BPD may benefit from visions that reconcile the psychologic with the physiologic mechanisms of the disorders [70].

### 4.3. An Evolutionary Outlook on BPD 

As a branch of behavioral ecology, Life History Theory (LHT) suggests that, within species, the individual allocation of resources to somatic growth/maintenance vs. reproduction depends on the unconscious individual prediction of future resource availability. In this regard, the Fast Life History Strategy (LHS) is described as the response to an early, unstable environment. In fast LHS, the trade-off between somatic growth and reproduction is oriented towards reproduction. As a consequence, fast LH individuals show faster development including earlier sexual maturation, earlier sexual activity, greater impulsivity, and increased risk-taking behavior [71], when compared to subjects that have been raised in safer environments. Through this lens, BPD subjects may be reflecting a pathological variant of the fast LH strategy [72,73].

From the study of a sample of more than 30,000 individuals, Baptista et al. recently confirmed that early life adversity was significantly higher in participants with BPD [74]. Also, participants with BPD had more children, earlier first sexual intercourse, a history of sexually transmitted disease, a higher BMI, and more metabolic risk factors. Thus, on a populational level, the emergence of BPD was confirmed to be associated with early adversity, and, later in life, with the prioritization of short-term reproductive goals over somatic maintenance. Overall, this conceptualization of BPD symptoms may promote the view that early forms of BPD are in fact an adaptive response to a complex, stressful, and unpredictable environment and that this adaptation may become dysfunctional and pathological in some individuals. This vision differs from the more traditional medical view of a mental disorder as a deficit caused by brain damage [72], may be particularly useful as a tool for a better understanding of the rationale underlying the disorder, and may help build more confidence, especially for adolescents. Also, this view highlights the importance of considering somatic interventions in the care of adolescents with BPD patients and puts special attention on early sexual behavior and its specific risks (sexually transmitted diseases, sexual abuse, unwanted early pregnancy, or parenthood) [75].

### 4.4. The Case for an Integrative Ecological Approach

Empirical evidence supporting the efficiency of specific psychotherapies for adolescent-onset BPD is limited. In contrast, clinical observations show the benefit of providing the patient and his family with a care network of intensive daily support for clinical and functional recovery [76]. This approach is in line with the concept of rehabilitation in adults, which is relatively more familiar to child and adolescent psychiatrists, as young patients are generally strongly embedded in multiple social networks, including families, schools, and other educational partners. The benefit of non-specific interventions for adolescents with BPD may partly explain why the difference between specific psychotherapeutic approaches and active control groups is difficult to determine. An ecological approach may be developed at different stages of the disorder and care. It may be introduced in hospital settings or included in day-treatment facilities or outpatient options. Interestingly, this enrichment of therapeutic partners (schools, educators) in the patient environment may also participate in the counter-transference aspects involved. The diffusion of the transference dynamics to multiple professionals could partly preclude the risk of care discontinuity due to excessive idealization and disqualification when only one therapist exists;Such an ecologically multilayered approach may be a chance not to focus just on health care and medical interventions. This aspect has been addressed recently in the field of general mental health in youth, which is frequently centered on medical aspects [77]. According to the authors, exclusive medicalization may undervalue parents and undermine all adults’ capacity to help. Social workers, teachers, relatives, parents, and adult peers should be fully associated with the rehabilitation process and empowered to do so;Finally, the therapeutic alliance with patients and families may be fragile, which may contribute to a particularly high attrition rate in controlled studies. Also, chronic care for these patients often implies the ability to adapt to daily situations. This flexibility may be life-saving but is also particularly difficult to assess through randomized controlled trials. Overall, we postulate that data emerging from clinical practice should not be dismissed.

## 5. Conclusions and Recommendations

Based on the recent finding review, the following clinical recommendations are proposed:Since adolescent-onset BPD is recognized as a valid and reliable clinical entity, the diagnosis should be provided by the clinicians and explained to both the patient and the family. When sub-threshold BPD is present, terms such as “BPD features” should be preferred. The natural history of BPD should be explained to the patient and family, in particular the main symptoms, prognosis, and discussion about realistic therapeutic objectives. The evolutionist narrative of BPD may be evoked;The high rate of comorbid psychiatric and neurodevelopmental disorders should be explored. Clinician attention should be paid to the distinction between vulnerability factors (e.g., attention deficit disorder, trauma-related disorder), subsequent emotional disorders, and pseudo-comorbidity;Early interventions should be prioritized with a proactive therapeutic position;Security plan for the management of life-threatening behaviors (e.g., self-harm, suicide attempts, risky behavior) should be anticipated based on crisis-based interventions (e.g., short hospitalizations, pharmacological treatment if needed);No pharmacological approach has been shown to be significantly effective on core BPD symptoms in adolescents, although comorbid psychiatric and neurodevelopmental disorders should be adequately addressed;Specific adolescent-adapted psychotherapy approaches exist, such as DBT or MBT, and are generally well-accepted. However, their specific efficacy is difficult to distinguish from the overall non-specific effect of integrative care. Multidisciplinary and collaborative networks, including family, social workers, and school environments, provide greater leverage than individual psychotherapy alone. The combination of different settings for sessions (e.g., individual, group therapists, telephone calls) showed a positive effect on care continuity and positively impacted the adolescent’s feeling of safety;Associate somatic-based interventions (e.g., relaxation, body-mind approaches) should be discussed in particular in the context of associated stress- and trauma-related disorders.

Regarding research implications, a better understanding of the global burden of the disease, including the social, educational, economic, and somatic health outcomes of adolescent-onset BPD, should be promoted, as well as informative epidemiological data based on large longitudinal cohorts. Specific research on preventive interventions should also be developed, focusing on low-cost, development-informed, and easy-to-implement ones. For example, specific programs of early detection/structured care may be implemented in foster care children, who are known to be at risk for ACEs. Complementary approaches to care should be evaluated for the at-risk group of adolescents. Relaxation techniques that aim to decrease the vagal tone may be of particular interest in this age group, characterized by its impulsiveness.

## Figures and Tables

**Figure 1 jcm-12-06668-f001:**
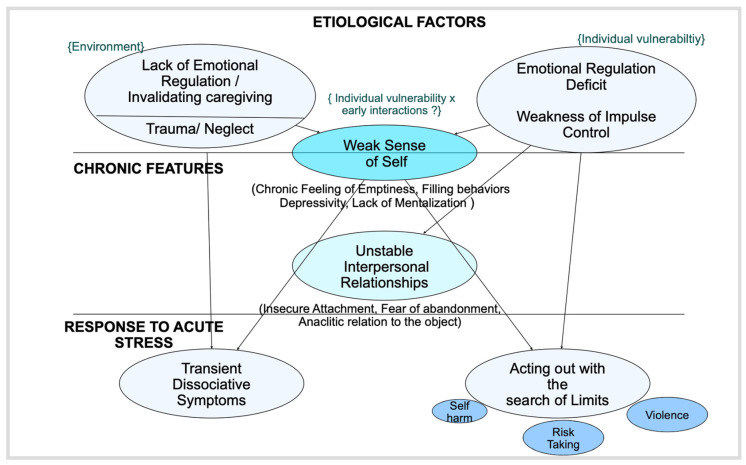
Developmental model of BPD.

## Data Availability

No new data were created or analyzed in this study. Data sharing is not applicable to this article.

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
