# Peer review of "Therapeutic and Preventive Interventions in Adolescents with Borderline Personality Disorder: Recent Findings, Current Challenges, and Future Directions"

_jcm, 2023, doi:10.3390/jcm12206668_

Round 1
Reviewer 1 Report
The manuscript is informative and of value. There are some minor concerns:
- The numbering of the references is duplicated.
- Some references are written in blue font, others in black font, please unify.
- The way of citing (list of references) is different than the one recommended by the journal, please check the instructions for authors and correct it.
- Please comment that BPD is closely associated with many structural and functional brain impairments, and thus, various forms of therapy can be ineffective. Therefore, it is important to describe and understand the neurobiological mechanisms of BPD in order to plan appropriate therapeutic interventions. Please refer for instance:
Gawda B., Bernacka R. Gawda A. (2016). The neural mechanisms underlying personality disorders. NeuroQuantology, 14 (2), 347-355.
Koenigsberg HW, Denny BT, Fan J, Liu X, Guerreri S, Mayson SJ, et al. (2014). The neural correlates of anomalous habituation to negative emotional pictures in borderline and avoidant personality disorder patients. Am J Psychiatry, 171(1), 82-90.
Reviewer 2 Report
**General Comments:**
1. **Clarity of Purpose**: The manuscript addresses an important and often overlooked topic in the field of psychiatry – borderline personality disorder in adolescents. However, the purpose of the review should be more explicitly stated in the introduction, along with the significance of the issue for clinicians and researchers.
2. **Organizational Structure**: The overall structure of the manuscript needs improvement. Consider breaking it down into distinct sections such as Introduction, Methods, Results, Discussion, and Conclusion to enhance clarity and flow.
3. **Literature Review**: The literature review should be expanded to provide a more comprehensive overview of the current state of research on borderline personality disorder in adolescents. Ensure that you have included recent and relevant studies.
**Specific Comments:**
1. **Treatment Recommendations**: Provide a more detailed discussion of the specific treatment recommendations for adolescents with borderline personality disorder. For instance, elaborate on the criteria for selecting pharmacological versus non-pharmacological interventions and their relative effectiveness.
2. **Psychotherapy Efficacy**: When discussing psychotherapy approaches like Dialectical Behavioral Therapy and Mentalization-Based Therapy, consider including empirical evidence that supports their efficacy in adolescent populations. Also, highlight any differences in outcomes compared to adult populations.
3. **Preventive Strategies**: Expand upon the preventive strategies for borderline personality disorder in adolescents. Elaborate on the specific interventions that have shown promise in reducing the risk factors related to early adversity.
4. **Research Gaps**: Identify and discuss gaps in the current research on borderline personality disorder in adolescents. Suggest areas where further investigation is needed to improve our understanding and treatment of the disorder in this age group.
5. **Strength of Evidence**: When discussing the evidence supporting various interventions, be sure to clearly indicate the strength of the evidence (e.g., high-certainty, low-certainty) and any limitations or potential biases in the studies reviewed.
6. **Recommendations for Future Research**: In the conclusion, provide concrete recommendations for future research directions in the field of borderline personality disorder in adolescents. This could include suggestions for study designs, specific populations to target, or emerging therapeutic modalities.
7. **Language and Clarity**: Review the manuscript for language clarity and readability. Ensure that complex concepts are explained concisely and that the writing style is accessible to a broad readership, including both clinicians and researchers.
8. **References**: Verify that all references are up-to-date and relevant to the topic. Ensure that you have cited key studies that support your arguments and recommendations.
9. **Tables and Figures**: Consider incorporating tables or figures to visually summarize key findings, treatment recommendations, or comparative data on interventions.
**Ethical Considerations:**
10. **Ethical Considerations**: Address ethical considerations related to the treatment of adolescents with borderline personality disorder, such as informed consent, patient confidentiality, and the potential for stigmatization.
**Overall, a comprehensive revision that addresses these points will significantly strengthen the manuscript and contribute to the field's understanding of borderline personality disorder in adolescents.**
Minor editing of English language required
